# An Improved YOLOv8 Network for Detecting Electric Pylons Based on Optical Satellite Image

**DOI:** 10.3390/s24124012

**Published:** 2024-06-20

**Authors:** Xin Chi, Yu Sun, Yingjun Zhao, Donghua Lu, Yan Gao, Yiting Zhang

**Affiliations:** 1Beijing Research Institute of Uranium Geology, Beijing 100029, China; 13664284913@163.com (X.C.); zhaoyingjun@briug.cn (Y.Z.); alexgreat@126.com (D.L.); qibuhuaiguine@163.com (Y.G.); bwfyzwy224@126.com (Y.Z.); 2National Key Laboratory of Uranium Resources Exploration-Mining and Nuclear Remote Sensing, Beijing 100029, China

**Keywords:** EP-YOLOv8, electric pylon, optical satellite image, object detection

## Abstract

Electric pylons are crucial components of power infrastructure, requiring accurate detection and identification for effective monitoring of transmission lines. This paper proposes an innovative model, the EP-YOLOv8 network, which incorporates new modules: the DSLSK-SPPF and EMS-Head. The DSLSK-SPPF module is designed to capture the surrounding features of electric pylons more effectively, enhancing the model’s adaptability to the complex shapes of these structures. The EMS-Head module enhances the model’s ability to capture fine details of electric pylons while maintaining a lightweight design. The EP-YOLOv8 network optimizes traditional YOLOv8n parameters, demonstrating a significant improvement in electric pylon detection accuracy with an average mAP@0.5 value of 95.5%. The effective detection of electric pylons by the EP-YOLOv8 demonstrates its ability to overcome the inefficiencies inherent in existing optical satellite image-based models, particularly those related to the unique characteristics of electric pylons. This improvement will significantly aid in monitoring the operational status and layout of power infrastructure, providing crucial insights for infrastructure management and maintenance.

## 1. Introduction

Electric pylons are a crucial part of power infrastructure, and their detection and identification are significant for transmission line inspection. Initially, the inspection of electric pylons relied mainly on manual labor, which is time-consuming and labor-intensive. In the early 21st century, the rapid development and adoption of unmanned aerial vehicle (UAV) technology [1,2,3,4] enabled UAV inspections of electric pylons, significantly improving work efficiency.

Scholars have researched the identification of electric pylons using UAV images, achieving positive results [5,6,7]. For example, Wang et al. [8] created an aerial photography dataset, compared the performance of Faster R-CNN and YOLOv3, and used deep learning to detect electric pylons. However, due to flight costs and operational efficiency, UAV inspection is limited in achieving rapid wide-area detection. Satellite imagery offers a more effective solution, with most researchers using radar satellite images [9,10,11,12]. Xie et al. [13] defined the linear polarization ratio and degree to detect vertical structures like electric pylons in hybrid mode. Tian et al. [14] used deep learning to recognize electric pylons in high-resolution SAR images by combining YOLOv2 and VGG, achieving better results than YOLOv2 alone. Matikainen et al. [15] introduced remote sensing methods for transmission line corridor surveys, using synthetic aperture radar (SAR), optical satellite imagery, optical aerial imagery, and thermal images, providing new ideas for electric pylon detection. However, radar images typically have low spatial resolution, limiting their ability to accurately analyze the shape and structure of targets. In contrast, optical images offer clearer feature details and simpler data processing. The effectiveness of using deep learning to detect vertically oriented elongated targets in high-resolution optical satellite images has been demonstrated with the development of artificial intelligence [16].

Some scholars have focused on electric pylon detection using optical satellite images. For instance, Tian et al. [17] introduced an enhanced rotating frame detector and a model utilizing high-resolution remote sensing images to accurately detect electric pylons and estimate necessary consumables. Yang et al. [18] developed an integrated learning strategy to classify the state of electric pylons, addressing the problem of state recognition in high-resolution remote sensing images under a deep learning framework. Huang et al. [19] proposed the SI-STD network, a slender object detection method based on shadow information, capable of handling both the pylon body and its shadow as dual labels for comprehensive detection. Huang et al. [20] also designed the STC-Det network, which treats shadows and corresponding electric pylons as two different types of targets and uses shadows as auxiliary information to supplement target torso features lost in imaging.

Detecting vertical structural targets, such as electric pylons, in optical satellite images is more complex than detecting targets with features concentrated in the horizontal direction. This complexity arises from variations in viewing angles, heights, and backgrounds. When the satellite viewing angle is relatively small, the imaging information of the target may not be obvious, and recognition relies primarily on shadow information. However, as the viewing angle increases, the imaging information becomes more prominent, but the complex spatial structures and diverse shapes can also pose challenges. Therefore, identifying such targets is a challenging task in remote sensing. Despite some progress, electric pylon detection still faces challenges due to the diversity of pylon structures, complex background environments, and the resolution limitations of optical satellite images. These factors often result in incomplete or inaccurate detection, highlighting the need for more targeted and effective detection methods. Previous methods either did not fully consider the specific characteristics of electric pylons or suffered from deficiencies in accuracy, speed, or network complexity. While some approaches introduced shadow information to improve accuracy, they often required remote sensing image parameters, such as the solar elevation angle and satellite viewing angle, which are difficult to obtain. This often led to an increased parameter count and computational complexity. Thus, there is a need for an improved network that not only effectively detects the characteristics of electric pylons but also is lightweight and capable of real-time detection.

The YOLO (You Only Look Once) series has continuously evolved to adapt to various detection tasks and scenarios. In this paper, we chose the YOLOv8n framework to optimize the model, aiming to further improve the accuracy and robustness of the algorithm. The specific improvements include:(1)The innovative DSLSK-SPPF module, an enhancement of the original SPPF module, incorporates the large selective kernel block (LSK block) with large kernel convolution. This advancement more effectively captures the prominent shadow and transmission line features associated with electric pylons, thereby improving the model’s capability to handle intricate backgrounds. Additionally, integrating dynamic snake convolution (DS-Conv) into the LSK block enhances the model’s adaptation to the complex shapes of electric pylons, which are influenced by imaging and environmental factors. Moreover, the spatial selection mechanism in the DSLSK-SPPF module significantly improves the accuracy of feature selection.(2)The head section of the original YOLOv8 model has been improved using efficient multi-scale convolution (EMS-Conv), which integrates multi-scale information fusion. This modification enhances the feature representation capability, improving the model’s ability to capture the details of electric pylons while maintaining its lightweight design.

Through these innovative modules and techniques, the mean average precision (mAP) of target detection reaches 95.5%, while significantly optimizing the detection accuracy and network parameters. This achievement realizes precise detection, providing a new method for power system inspection. Additionally, it also offers a promising technological foundation for similar vertical structure target detection scenarios, such as wind turbines.

## 2. The Architecture of YOLOv8

YOLOv8 is an efficient one-stage object detection method that eliminates the need for anchor boxes. It offers significant advantages in both detection accuracy and speed [21]. By directly predicting bounding boxes within the grid, it circumvents the need for initializing anchor box sizes and performing non-maximum suppression calculations, thereby greatly enhancing object localization efficiency. This algorithm comes in five variants—n, s, m, l, and x—based on depth and width, catering to various scene requirements. However, as accuracy improves, there is a significant increase in model parameters and computational complexity.

YOLOv8 comprises an input end, backbone network, neck section, and head section. The input section performs adaptive anchor box calculations, adaptive grayscale padding, and mosaic data augmentation on input images to enhance the model’s robustness. The backbone network consists of CBS modules, C2f modules, and spatial pyramid pooling faster (SPPF) modules. Multiple convolution and C2f modules process input images to extract feature maps at different scales. The CBS module consists of convolution (Conv), batch normalization (BN), and sigmoid linear unit (SiLU) activation functions. Compared to the C3 module, the C2f module employs skip connections and additional split operations as the primary residual learning modules, maintaining lightweight characteristics while capturing richer gradient flow information. SPPF enhances the network’s receptive field, combines pooling and convolution operations for feature fusion, and adaptively integrates features of various scales, thereby improving object detection effectiveness and enhancing the model’s feature extraction capability.

The neck section combines the feature pyramid network (FPN) [22] and the path aggregation network (PAN) [23] through top-down and bottom-up cross-layer connections. This enables more comprehensive feature fusion, compensating for feature loss caused by network deepening and enriching feature granularity.

The head section uses a decoupled head structure, separating detection from classification. Scores obtained by weighting classification and regression scores determine positive and negative samples, effectively improving model performance. The YOLOv8 network architecture is illustrated in Figure 1.

## 3. Improved YOLOv8 Network

Compared to two-stage detection algorithms with high computational complexity and complex model structures, as well as other single-stage algorithms, YOLO stands out for its fast execution speed, high accuracy, and strong generalization ability. The YOLO series has become indispensable in the field of object detection, consistently improving in both detection speed and accuracy. These algorithms continuously evolve in areas like data augmentation, backbone networks, feature fusion, and detection subnetworks to adapt to various tasks and scenarios. YOLOv8 builds upon the success of previous YOLO versions, offering comprehensive improvements and innovations to enhance performance and flexibility [24]. In this paper, we further optimize the model by integrating the YOLOv8 framework to enhance algorithmic accuracy.

However, in practical detection processes, the standard convolution used in the YOLOv8 network is not ideal for extracting features of electric pylon targets. The complex background environments significantly interfere with electric pylon detection. Additionally, factors like climate, surface environment, time, and shooting angles affect the size, accuracy, and clarity of electric pylons. These factors result in less than satisfactory performance of the YOLOv8 network in electric pylon detection [25].

### 3.1. Optimization of Spatial Pyramid Pooling Faster

To obtain richer multi-scale feature representations and improve the detection accuracy and speed of the algorithm, the SPPF module [26] integrates information from multi-scale local features. This module provides additional information from a wider range of spatial levels [27], giving the network a global perspective. The SPPF module only needs to specify a convolutional kernel, and each pooling operation’s output serves as the input for the next, speeding up data processing [28]. This effectively reduces the computational load and improves the efficiency of feature extraction. By combining local and global features, the SPPF module extends the network’s receptive field and fuses multi-scale features in a cascaded manner, enhancing object detection performance, as illustrated in Figure 2.

However, the original SPPF module’s fixed-size grids for pooling operations may not be optimal for objects of varying sizes and scales. Additionally, SPPF relies on a concatenation of three max-pooling operations to extract features from the input. Max-pooling captures only the maximum values, reflecting local features [29] and potentially missing crucial contextual information. This limitation is significant for detecting electric pylons, often accompanied by prominent shadow and transmission line features. To address these challenges, this paper introduces the LSK block attention mechanism into the SPPF module, significantly enhancing its ability to manage the complex backgrounds typically found around electric pylons.

Due to the influence of various imaging and environmental factors, electric pylons’ shapes in satellite images often exhibit complexity and variability. To address this, we integrate DS-Conv into the LSK block, allowing the network’s receptive field to dynamically adjust. This integration significantly enhances the model’s capability to extract and emphasize the key features of electric pylons.

This dual consideration ensures that the DSLSK-SPPF module can effectively leverage both the surrounding features and the distinctive characteristics of the pylons themselves, achieving superior detection performance.

The large selective kernel network (LSKNet) [30] is a network architecture designed specifically for remote sensing object detection. It considers prior knowledge around objects in remote sensing images and can dynamically adjust its larger receptive field through a spatial selection mechanism to understand prior knowledge in the background. LSKNet, inspired by ConvNeXt [31], proposes the LSK block structure, which is a repeating block in the backbone network, as shown in Figure 3.

Each LSKNet block consists of two residual sub-blocks: the large kernel selection (LKSelection) sub-block and the feed-forward network (FFN) sub-block. The LKSelection sub-block includes fully connected (FC) layers, large selective kernel (LSK) modules, and GELU activation functions. The LSK module (shown in Figure 4) consists of a series of large kernel convolutions and a spatial selection mechanism. By dynamically adjusting its receptive field and dynamically selecting appropriate convolutional kernels, the LSK module adapts to different object types and backgrounds, as well as various types of contextual information. The spatial selection mechanism is an adaptive weight allocation method that enhances attention to the most relevant spatial regions of the target, ultimately improving the accuracy of object detection. The FFN sub-block is used for channel mixing and feature refinement, comprising a sequence of FC layers, depth-wise convolutions (DW-Conv), and GELU activation functions.

The spatial selection mechanism helps the network better understand and capture spatial information relevant to the target by selecting feature maps from different scales. It first concatenates features U obtained from different convolutional kernels with different receptive field ranges. Then, it effectively extracts spatial relationships by applying channel-wise average pooling and max-pooling:(1)$${SA}_{avg}=P_{avg}\left(U\right),\;{SA}_{avg}=P_{avg}\left(U\right)$$

Next, through a convolutional layer $F^{2\to N}$, the merged feature tensor is transformed into N spatial attention maps. For each spatial attention map, a binary tensor representing the spatial selection mask is obtained through the sigmoid activation function. Then, the feature tensor processed by a series of decomposed large convolutional kernels is weighted summed based on the corresponding spatial selection mask, and finally fused through another convolutional layer F to obtain an attention feature tensor S:(2)$$S=F(\sum_{i=1}^{N}\left({SA}_{i}\cdot U_{i}\right))$$

The LSK module produces its final output feature Y by performing element-wise multiplication between the input feature X and the attention feature tensor S:(3)Y=X·S

However, the fixed shape of the standard convolutional kernel results in a limited receptive field, restricting the model’s ability to extract features from objects of various sizes and shapes. This limitation is particularly problematic for detecting electric pylons, which often have complex structures. To address this issue and enhance the feature extraction capability of the improved SPPF module, as well as its ability to adapt to the complex shapes of electric pylons, this paper introduces the DS-Conv in the redesign of the DSLSK-SPPF module, as shown in Figure 5. DS-Conv allows the network to dynamically adjust its receptive field to accommodate the irregular yet generally elongated shapes of electric pylons in images. This improvement significantly enhances the model’s ability to extract relevant features from electric pylons of different scales, resulting in increased detection accuracy and robustness. Combining the LSK block with DS-Conv specifically addresses the challenges posed by the unique characteristics of electric pylons, highlighting the method’s effectiveness.

Deformable convolution introduces deformation offsets, allowing the convolution kernels to flexibly focus on the complex geometric features of the target, as shown in Figure 6. However, if the model learns deformable offsets freely, the receptive field tends to deviate from the target, especially in the case of elongated structures. Therefore, Qi et al. [32] adopt an iterative strategy to sequentially select observed positions for each target to ensure the continuity of attention. Due to the large deformation offsets, the perception field will not spread too far. DS-Conv is an innovative convolution method aimed at capturing complex geometric features in images with higher precision, as shown in Figure 7.

DS-Conv dynamically adapts to the shape of the target by adjusting the position of the convolution kernel along the *x*-axis and *y*-axis. Taking the *x*-axis direction as an example, for a convolution kernel of size 9, the specific position of each grid point is represented as $K_{i\pm c}=(x_{i\pm c},y_{i\pm c})$, where c represents the horizontal distance from the central grid. The selection of each grid point is an accumulative process, starting from the central position $K_{i}$, and the positions away from the central grid depend on the position of the previous grid. This expression of displacement along the *x*-axis ensures that the convolution kernel maintains a linear structure without deviating from the main contour of the target. Similarly, similar adjustments are made along the *y*-axis to ensure more accurate matching of the convolution kernel to the target. Here, Σ denotes cumulative deformation:(4)$$K_{i\pm c}=\left\{\begin{array}{c} {(x}_{i+c},y_{i+c})=\left(x_{i}+c,y_{i}+\sum_{i}^{i+c}∆y\right)\\ {(x}_{i-c},y_{i-c})=\left(x_{i}-c,y_{i}+\sum_{i-c}^{i}∆y\right) \end{array}\right.$$
(5)$$K_{j\pm c}=\left\{\begin{array}{c} {(x}_{j+c},y_{j+c})=\left(x_{j}+\sum_{j}^{j+c}∆x,y_{i}+c\right)\\ {(x}_{j-c},y_{j-c})=\left(x_{j}+\sum_{j-c}^{j}∆x,y_{i}-c\right) \end{array}\right.$$

Since the offsets are typically decimals, bilinear interpolation is used to compute precise positions:(6)$$K=\sum_{K\prime }B\left(K^{\prime },K\right)\cdot K\prime$$
where K denotes the fractional position of the convolution kernel along the coordinate axis, K′ enumerates all integer spatial positions, and B is the bilinear interpolation kernel. Bilinear interpolation divides the computation of the convolution kernel into two one-dimensional convolution kernels, corresponding to the interpolation along the *x*-axis and *y*-axis directions, respectively.
(7)$$B\left(K^{\prime },K\right)=b\left(K_{x},{K\prime }_{x}\right)\cdot b\left(K_{y},{K\prime }_{y}\right)$$

DS-Conv covers a range of 9 × 9 during the deformation process, adapting better to elongated structures through two-dimensional (*x*-axis, *y*-axis) changes, thereby better perceiving key features.

In optimizing the SPPF, integrating the LSK block with DS-Conv significantly enhances the module’s ability to utilize surrounding information, such as shadows and transmission line features, in addition to the complex characteristics of the electric pylons themselves. The redesigned DSLSK-SPPF module benefits from a spatial selection mechanism, which allows for more effective feature extraction. Consequently, the model demonstrates improved detection accuracy and robustness in identifying electric pylons across various scales and conditions.

### 3.2. Optimization of Head

The head section of YOLOv8 has a low parameter count and limited expressiveness, making it difficult to deeply explore spatial structural information. Additionally, its single-scale prediction structure and lack of dynamic learning capability pose certain limitations on object detection. Electric pylons have complex spatial structures and diverse shapes, posing higher requirements for object detection models. Traditional single-scale prediction structures are insufficient for effectively capturing these details.

To address these limitations, this paper introduces the EMS-Conv method to improve the YOLOv8 head. EMS-Conv enhances feature representation by integrating multi-scale features through convolutions at different scales. This approach enables the model to capture both global structures and local details of electric pylons, thereby improving detection accuracy.

In traditional convolutional neural networks (CNNs), the use of single-scale convolutional kernels restricts the network’s ability to effectively capture features at different scales, thereby hindering its recognition of objects or scenes with different scales. Multi-scale convolution techniques better capture features at different scales and extract richer information. Among them, FPN is a typical multi-scale convolution technique that fuses features obtained at different scales through bottom-up and top-down methods via lateral connections. Inspired by FPN, Sun et al. [33] proposed the efficient multi-scale convolution (EMS-Conv) module to achieve multi-scale information fusion. The workflow of the EMS-Conv module is as follows: firstly, the inputs obtained from the convolution are divided into two groups based on the number of channels, with the first group processing the original image to extract base and untreated features. Then, the second group further divides the untreated features into two parts and convolves them separately using 3 × 3 and 5 × 5 convolution kernels. The results of this operation are then concatenated with the base features obtained from the first group, forming independently connected features in each channel. Finally, a 1 × 1 convolution layer is added after the concatenation output to fuse channel information, achieving an improvement in feature expression capability and dimensionality reduction, as shown in Figure 8.

This paper adopts the concept of parameter sharing [34,35] and incorporates EMS-Conv to enhance the original convolution module in the head section of YOLOv8, as illustrated in Figure 9. Upon receiving input in the head, an EMS-Conv shared parameter module delays the branches before proceeding with decoupled classification and regression.

Through this improvement, the model enhances the accuracy of electric pylon detection, providing a new method for detecting complex structural targets while ensuring the lightweight design of the head component.

## 4. Results and Analysis

### 4.1. Experimental Environment

The experiments were conducted on a mobile server equipped with an NVIDIA A100-SXM4 GPU (NVIDIA, Santa Clara, CA, USA) and an Intel(R) Xeon(R) Gold 6326 CPU (Intel, Santa Clara, CA, USA). Model training was performed using the CUDA 12.2 configuration and the PyTorch 2.0.0 deep learning framework, with YOLOv8n as the baseline network model.

During the training phase, a learning rate decay method was employed to optimize model performance. This method adjusts the update speed of model parameters using an initial learning rate (lr0) and a learning rate coefficient (lrf) to control the decay of the learning rate during training. The final learning rate was determined by multiplying the initial learning rate by the coefficient. To ensure an adequate number of training steps, the iteration count was set to 200 epochs. Table 1 shows the hyperparameters used during the training process.

### 4.2. Dataset and Evaluation Metrics

In this paper, we utilized the open-source EPD dataset [16], a collection of high-resolution optical satellite images specifically designed for electric pylon detection. The dataset comprises 1500 images, with 720 images captured along the main power grid in Guangdong Province by the Pleiades satellite, while the remaining images were sourced from Google Earth. The Pleiades satellite images are orthorectified, while the Google Earth images consist of multispectral products captured by different sensors. The EPD dataset includes over 3000 electric pylons and features images with complex scenarios, such as varied background colors and interfering objects. This diversity helps in better assessing the performance of pylon detection under challenging conditions. Each image in the EPD dataset has a size of 1024 × 1024 pixels and a spatial resolution of 1 m. The images were partitioned into training, validation, and test sets in an 8:1:1 ratio.

The experimental results in this paper were evaluated using several commonly used metrics in object detection: precision (P) reflects the model’s ability to correctly identify relevant objects, while recall (R) measures the model’s capacity to identify all pertinent objects. The mAP is used to assess the overall accuracy of the model. GFLOPs, which stands for giga floating-point operations per second, quantifies the computational complexity of the network model, enabling a fair comparison of detection speeds among different algorithms. The parameters (Para) denotes the total number of parameters required for model training, evaluating the size and complexity of the model. Lastly, frames per second (FPS) indicates the number of frames processed per second.

The calculation formulas for precision and recall are as follows:(8)$$P=\frac{TP}{TP+FP}$$
(9)$$R=\frac{TP}{TP+FN}$$
where TP represents the number of samples correctly classified as positive, FP represents the number of samples falsely classified as positive, and FN represents the number of samples falsely classified as negative. Subsequently, mAP is calculated as:(10)$$mAP=\frac{1}{N}\sum_{i=1}^{N}{AP}_{i}$$
where N represents the total number of categories, AP represents the area under the precision-recall curve for a specific category at different confidence levels, and mAP@0.5 refers to the mean average precision at a 0.5 threshold for the intersection over union (IoU) of predicted boxes with true objects.

### 4.3. Performance Comparison of Different Spatial Pyramid Pooling

This paper explores and compares the influence of different spatial pyramid pooling (SPP) layer structures on model performance. Table 2 presents the impact of various pooling layers (including SPP [36], SPPF, SPPCSPC [37], SPPFCSPC [38], and DSLSK-SPPF) on accuracy, precision, recall, and other performance metrics. Compared to the original SPPF module in YOLOv8, DSLSK-SPPF significantly increased accuracy by 3.7%, showing improvements over other spatial pyramid pooling layers. DSLSK-SPPF achieved the highest precision at 94.3%. While DSLSK-SPPF had a slightly lower recall compared to some other models like SPPFCSPC, its recall remained competitive, demonstrating a strong ability to detect objects even if not the highest. Although DSLSK-SPPF had the lowest FPS at 116 frames per second, this is a reasonable trade-off for the substantial accuracy improvements it offers. The increase in parameters and computational load is acceptable due to the significant accuracy improvement, which is crucial for enhancing object detection performance.

### 4.4. Ablation Experiment

To better validate the impact of the improved modules and their combinations on the performance of the original model, we designed ablation experiments. These experiments compared the improved modules against the YOLOv8n model, which served as the baseline. The experimental results presented in Table 3 demonstrate that incorporating various improved modules enhanced the parameters, GFLOPs, and detection accuracy of the original network model. The initial model achieved a detection accuracy (mAP@0.5) of 91.2%, with a precision of 89.0% and a recall of 88.3%. When integrating the DSLSK-SPPF module, the detection accuracy improved to 94.9%, with precision increasing to 94.3% and recall to 88.7%. However, this integration significantly increased parameters and slightly decreased GFLOPs and FPS. Introducing the EMS-Head module further resulted in a detection accuracy of 94.3%, precision of 94.2%, and recall of 89.0%, while the GFLOPs and parameter count decreased significantly, and FPS improved substantially. Both the DSLSK-SPPF and EMS-Head modules individually contributed significantly to the precision and recall of object detection. Their combined effect, however, was even more pronounced. After simultaneously introducing both modules, the model achieved the highest detection accuracy of 95.5% on the mAP@0.5 metric, with precision at 93.4% and recall at 92.0%. This represents a 4.3% improvement over the original YOLOv8n network accuracy. However, these two modules slightly increased the computational burden and parameter count, resulting in a slight decrease in FPS. They enhanced the detection accuracy, precision, and recall in practical applications, thereby improving the model’s practicality and robustness.

The analysis presented in Figure 10 demonstrates that the EP-YOLOv8 model exhibits a significant improvement over the original YOLOv8n model in terms of the average mAP@0.5 value. By comparing the performance of the two models, it is evident that the EP-YOLOv8 model can more effectively detect and recognize targets, providing a more reliable foundation for applications.

To compare the performance of the original YOLOv8n model and the improved EP-YOLOv8 model in visual detection, experiments were conducted using the same set of images for object detection. The results clearly demonstrate a significant enhancement of the EP-YOLOv8 model relative to the original model. Specifically, the EP-YOLOv8 model effectively reduces instances of false positives and false negatives, thereby greatly improving the detection performance of electric pylons.

To perform comparative analysis, electric pylon images of different environmental backgrounds and scales were selected from the EPD dataset for detection and recognition. The detection results of the two models are shown in Figure 11, indicating varying degrees of false negatives with the YOLOv8n model, particularly in more complex background conditions. In some images, although the YOLOv8n model successfully detected all targets, false positive detections were also observed, along with inaccurate and overlapping bounding box predictions. In contrast, the proposed EP-YOLOv8 model demonstrated higher detection accuracy, lower false negative rates, and lower false positive rates across various scenes, with more accurate bounding box predictions. This confirms the superior performance of the EP-YOLOv8 model in various scenarios, significantly enhancing object detection effectiveness.

### 4.5. Performance Comparison of Different Models

To assess the enhanced model’s performance, this paper presents a series of comparative experiments against various widely employed object detection models. These models include two-stage anchor-based methodologies, represented by Faster R-CNN [39], and one-stage anchor-based approaches like SSD [40], YOLOv3 [41], and YOLOv5, along with their variants. The evaluation also incorporated the lightweight variant YOLOv3-tiny [42]. Furthermore, it extended to end-to-end object detection methods such as YOLOv10 [43] and RT-DETR [44], offering a comprehensive comparison across different detection paradigms. For RT-DETR, we specifically chose a lighter variant by equipping it with ResNet18 for our evaluations. These networks were selected not only for their widespread use in object detection tasks but also as benchmarks for improvements in electric pylon detection methods.

Table 4 demonstrates that EP-YOLOv8 exhibits significant improvements and excels in several key metrics. Firstly, EP-YOLOv8 achieves a precision of 93.4%, second only to YOLOv5s and YOLOv3, indicating strong detection accuracy. Additionally, it achieves the highest recall at 92.0% among all models, highlighting its superior ability to detect all targets. Moreover, EP-YOLOv8 attains an outstanding mAP@0.5 of 95.5%, significantly surpassing all other models, demonstrating its overall detection performance excellence. In terms of computational complexity, EP-YOLOv8’s GFLOPs is 7.7, matching YOLOv5n, indicating a lower computational resource requirement. With 6.3 million parameters, EP-YOLOv8, while not as lightweight as YOLOv5n and YOLOv10n, remains relatively compact, making it suitable for deployment in resource-constrained environments. Although EP-YOLOv8 has a lower processing speed compared to some models, it still maintains a relatively high level. Overall, EP-YOLOv8’s exceptional recall and mAP@0.5, combined with its moderate computational complexity and parameter count, make it an effective and accurate model for object detection tasks. It excellently balances efficiency and precision.

The EPD dataset, with a 1 m resolution, can capture intricate details of electric pylons, including supporting structures and crossarms. These fine details are crucial for accurate detection. To thoroughly evaluate the impact of resolution on detection performance, but constrained by data resources and electric pylon specifications, we down-sampled the original EPD dataset to a 2 m resolution using bilinear interpolation. We then conducted training and testing on this lower-resolution dataset. The results, detailed in Table 5, reveal that most models exhibit a slight decline in precision, recall, and mAP@0.5 after down-sampling to 2 m. This decline is expected, as down-sampling reduces image details, making target detection more challenging. Despite the performance drop in most models due to the resolution reduction, EP-YOLOv8 maintained a high mAP@0.5 of 94.5% at the 2 m resolution. This indicates that the EP-YOLOv8 model is highly adaptable to different resolutions, enabling it to intricately capture the structural complexities of power pylons. Moreover, it effectively integrates surrounding contextual information to enhance detection accuracy. This stability and adaptability in complex scenarios further underscore the model’s robustness and efficacy.

## 5. Discussion

Compared to widely used algorithms for electric pylon detection, the EP-YOLOv8 model demonstrates higher accuracy. However, challenges persist with false positives and negatives, primarily stemming from complex backgrounds characterized by dense vegetation, varied terrain, and occlusions. Although electric pylons exhibit distinct texture features and detection cues in images, they can blend with complex backgrounds, affecting detection results. Despite advancements in detection technology, EP-YOLOv8 still struggles with small, camouflaged targets. Further research is needed to address these issues and boost performance.

Additionally, while EP-YOLOv8 achieves higher accuracy and computational efficiency, this improvement comes at the cost of increased inference time and parameters. The DSLSK-SPPF module improves accuracy but adds complexity due to large kernel convolutions. To address this, a lightweight head section and EMS-Conv module were introduced, resulting in faster detection while maintaining accuracy. However, increasing inference speed remains a challenge, necessitating future research to optimize the performance of electric pylon detection. Techniques like data augmentation and network enhancements can improve model generalization and adaptability across diverse scenarios. Moreover, evaluating EP-YOLOv8 on larger and more diverse datasets will provide deeper insights into its robustness and reliability.

## 6. Conclusions

The detection and recognition of electric pylons are crucial steps in monitoring their operational status and are essential components of power line monitoring. In this paper, an innovative electric pylon target detection model, EP-YOLOv8, is proposed based on the YOLOv8n network. Compared to traditional methods, EP-YOLOv8 has a distinct advantage in effectively adapting to the specific features of electric pylons and establishing a closer connection with the target. This advantage is primarily due to the enhancement of the SPPF in the original network. The integration of the LSK block within the innovative DSLSK-SPPF module significantly improves its efficiency in capturing detailed information in remote sensing scenes. Furthermore, the utilization of DS-Conv allows dynamic adaptation to irregular focal lengths within the image. These improvements enable the DSLSK-SPPF to simultaneously consider the complex shapes of the targets and the surrounding information that aids in detection.

Simultaneously, EMS-Head allows EP-YOLOv8 to maintain a lightweight design while enhancing its ability to capture the intricate details of electric pylons. These advancements empower EP-YOLOv8 with higher accuracy and enhanced adaptability in electric pylon target detection tasks, effectively meeting the demands of pylon monitoring.

Experimental results demonstrate that the EP-YOLOv8 model exhibits excellent performance in detecting electric pylons in scenes with complex backgrounds and multiple scales. Compared to other mainstream object detection models such as Faster R-CNN, SSD, YOLOv3, YOLOv3-tiny, YOLOv5n, YOLOv5s, YOLOv8n, YOLOv8s, YOLOv10n, YOLOv10s, and RT-DETR-R18, EP-YOLOv8 improves detection accuracy on the EPD dataset by 3.0% to 14.9%. Furthermore, the EP-YOLOv8 model not only enhances detection accuracy but also reduces computational requirements, demonstrating high efficiency. Future research directions may include supplementing potentially lost features during the imaging process, such as shadows, by introducing external information, while ensuring real-time monitoring, thereby further improving the accuracy and practicality of electric pylon target detection and classification. There are also plans to extend the EP-YOLOv8 model to other scenarios similar to electric pylons, thereby expanding the application scope of vertical structure target detection in satellite imagery.

## Figures and Tables

**Figure 1 sensors-24-04012-f001:**
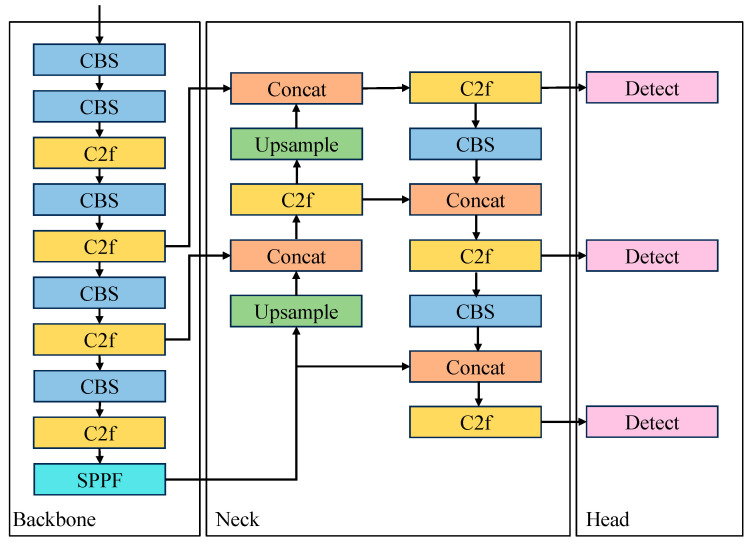
The architecture of YOLOv8 network.

**Figure 2 sensors-24-04012-f002:**
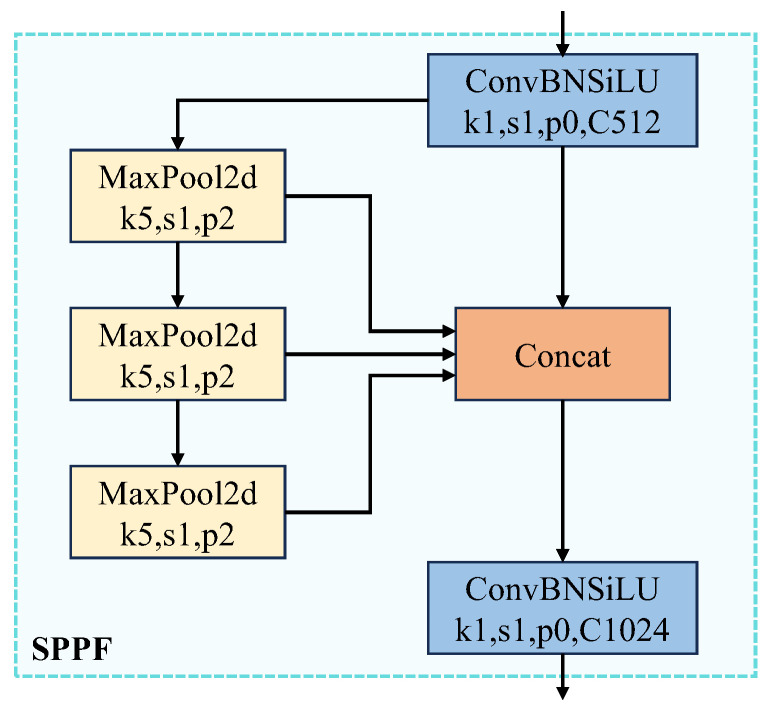
Schematic diagram of the SPPF module structure.

**Figure 3 sensors-24-04012-f003:**
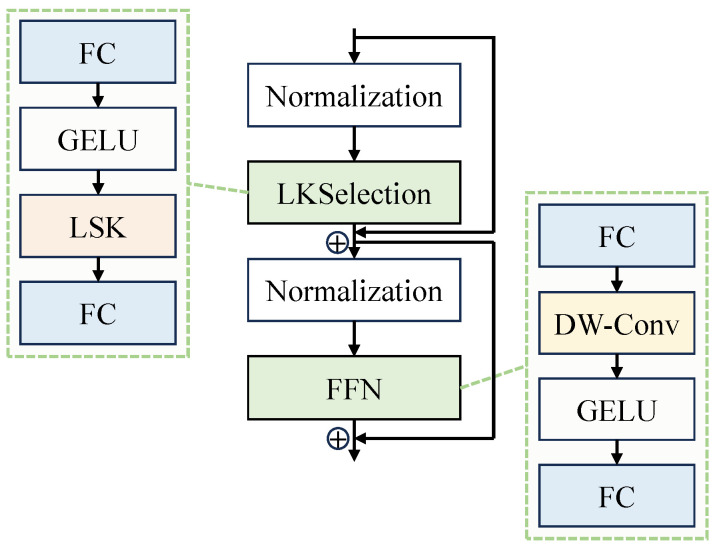
Schematic diagram of the LSK block module.

**Figure 4 sensors-24-04012-f004:**
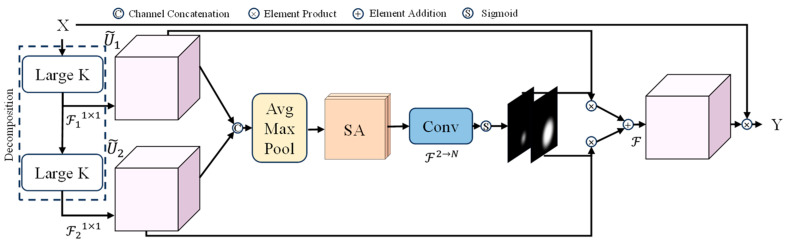
Schematic diagram of the LSK module.

**Figure 5 sensors-24-04012-f005:**
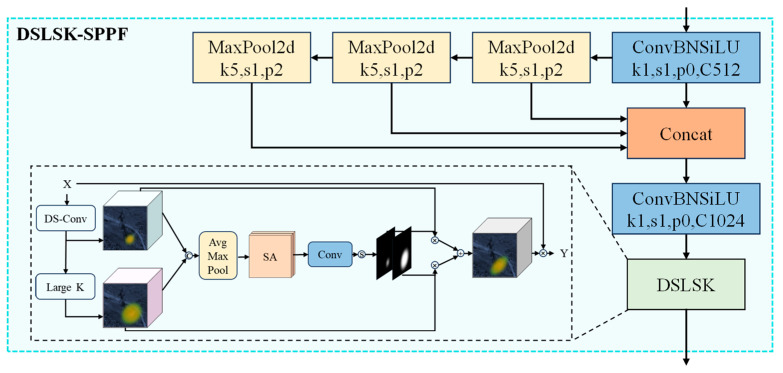
Schematic diagram of the DSLSK-SPPF module.

**Figure 6 sensors-24-04012-f006:**
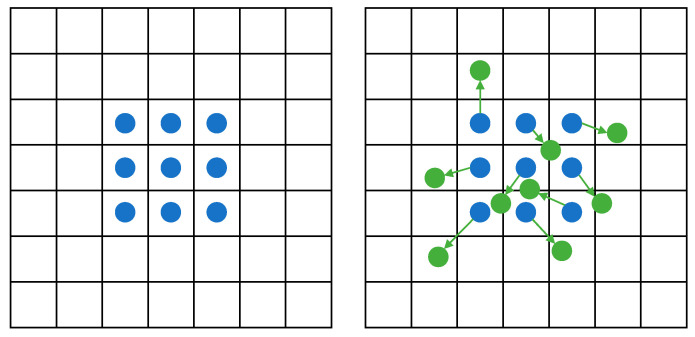
Sampling methods of regular convolution and deformable convolution Blue dots represent standard convolutions, while green dots represent deformable convolutions (Offset).

**Figure 7 sensors-24-04012-f007:**
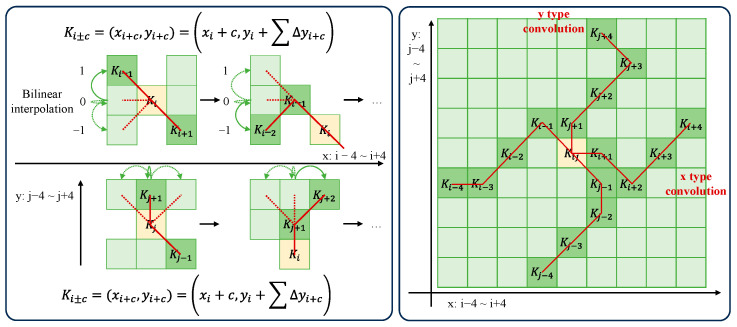
Schematic diagram illustrating the computation of dynamic snake convolution kernel coordinates and optional receptive fields.

**Figure 8 sensors-24-04012-f008:**
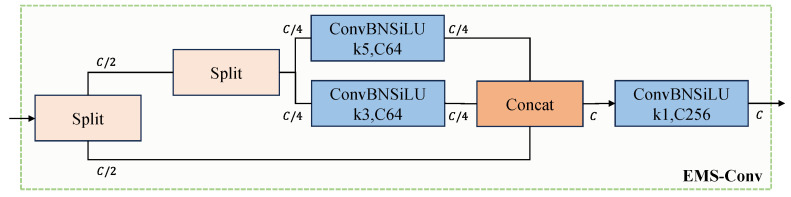
Schematic diagram of the EMS-Conv module.

**Figure 9 sensors-24-04012-f009:**
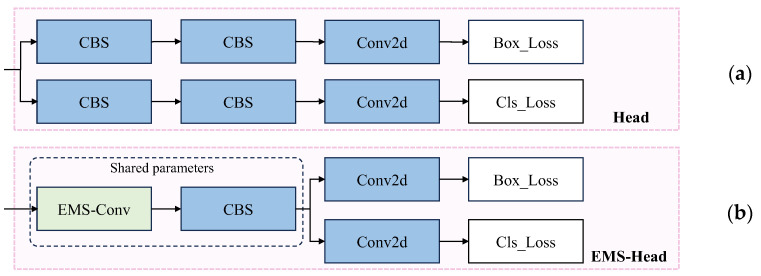
Comparison of the detector head: (**a**) YOLOv8 detection head structure; (**b**) reconstructed detection head structure.

**Figure 10 sensors-24-04012-f010:**
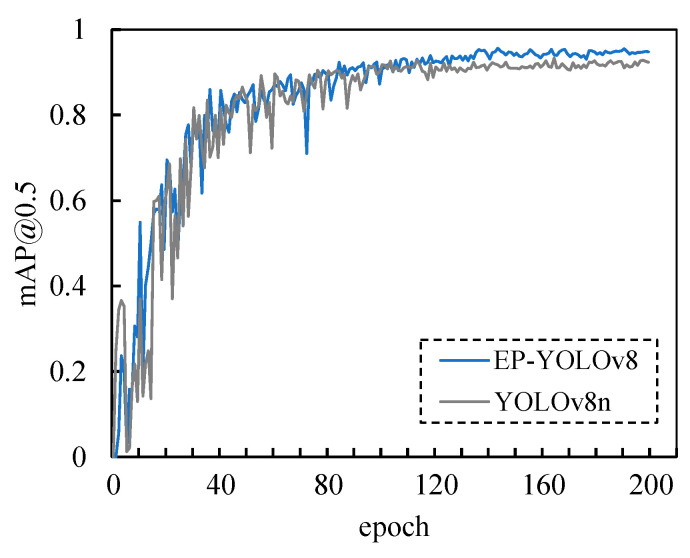
Comparison of mAP@0.5 values between EP-YOLOv8 model and original model.

**Figure 11 sensors-24-04012-f011:**
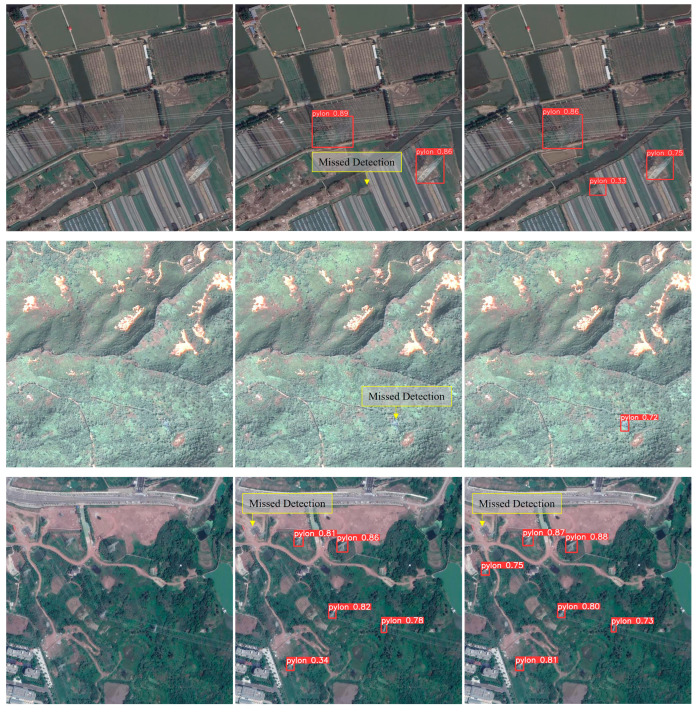
Comparison of detection performance: (**a**) input image; (**b**) object detection result of YOLOv8n; (**c**) object detection result of EP-YOLOv8.

**Table 1 sensors-24-04012-t001:** Hyperparametric configuration.

Hyperparameter Options	Setting
Input Resolution	640 × 640
Initial Learning Rate 0 (lr0)	0.01
Learning Rate Float (lrf)	0.001
Batch_size	64
Epochs	200

**Table 2 sensors-24-04012-t002:** Performance table of different spatial pyramid pooling.

Models	P (%)	R (%)	mAP@0.5 (%)	GFLOPs	Para (M)	FPS/(Frames − 1)
YOLOv8n + SPP	91.7	89.0	93.8	8.7	3.2	141
YOLOv8n + SPPF	89.0	88.3	91.2	8.7	3.2	146
YOLOv8n + SPPCSPC	93.2	89.7	93.6	10.0	4.8	118
YOLOv8n + SPPFCSPC	90.3	89.8	93.5	10.0	4.8	123
YOLOv8n + DSLSK-SPPF	94.3	88.7	94.9	9.9	6.6	116

**Table 3 sensors-24-04012-t003:** Ablation experiments with the modules.

Baseline	DSLSK-SPPF	EMS-Head	P (%)	R (%)	mAP@0.5 (%)	GFLOPs	Para (M)	FPS/(Frames − 1)
YOLO v8n			89.0	88.3	91.2	8.7	3.2	146
	√		94.3	88.7	94.9	9.9	6.6	116
		√	94.2	89.0	94.3	5.9	2.6	156
	√	√	93.4	92.0	95.5	7.7	6.3	111

**Table 4 sensors-24-04012-t004:** Performance table of different models in detection. Train resolution, 1 m/pixel; test resolution, 1 m/pixel.

Models	P (%)	R (%)	mAP@0.5(%)	GFLOPs	Para(M)	FPS/(Frames − 1)
Faster R-CNN	87.1	91.7	90.8	941.0	28.3	11
SSD	85.1	74.7	80.6	62.7	26.3	76
YOLOv3	95.3	83.7	92.3	282.6	103.7	119
YOLOv3-tiny	92.9	83.4	90.9	18.9	12.1	222
YOLOv5n	90.3	85.0	91.5	7.7	2.7	145
YOLOv5s	96.1	84.4	92.7	24.0	9.1	126
YOLOv8n	89.0	88.3	91.2	8.7	3.2	146
YOLOv8s	93.0	88.2	92.5	33.7	17.6	128
YOLOv10n	94.4	83.5	91.2	8.2	2.7	113
YOLOv10s	91.7	89.4	92.3	24.4	8.0	110
RT-DETR-R18	94.0	84.3	92.6	56.9	19.9	86
EP-YOLOv8	93.4	92.0	95.5	7.7	6.3	111

**Table 5 sensors-24-04012-t005:** Performance table of different models in detection. Train resolution, 2 m/pixel; test resolution, 2 m/pixel.

Models	P (%)	R (%)	mAP@0.5(%)
Faster R-CNN	82.0	88.9	86.9
SSD	84.4	76.0	79.9
YOLOv3	88.6	88.7	91.8
YOLOv3-tiny	90.1	81.7	88.4
YOLOv5n	92.3	79.2	89.3
YOLOv5s	94.5	85.5	92.2
YOLOv8n	89.8	81.5	88.4
YOLOv8s	90.7	86.4	91.6
YOLOv10n	85.8	86.0	90.4
YOLOv10s	91.2	87.0	90.3
RT-DETR-R18	89.8	82.2	90.0
EP-YOLOv8	92.8	89.4	94.5

## Data Availability

The dataset is sourced from EDP, and it can be downloaded from the following website: https://universe.roboflow.com/robin-public/electric-pylon-detection-in-rsi (accessed on 24 December 2022).

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
