# Peer review of "An Improved YOLOv8 Network for Detecting Electric Pylons Based on Optical Satellite Image"

_sensors, 2024, doi:10.3390/s24124012_

Round 1

Reviewer 1 Report

Comments and Suggestions for Authors

The authors propose a fusion network named an EP-YOLOv8 network by integrating dynamic snake convolution, large selective kernel network block, and efficient multi-scale convolution into the YOLOv8n model. The authors conduct a series of experiments to verify the effectiveness of the proposed method. However, there are several issues that should be considered by the authors.

1) The review of electric pylons detection is not sufficient and needs to be improved.

2) The dynamic snake convolution and large selective kernel network block utilized in this method have been applied in many fields, lacking a high degree of innovation.

3) An in-depth explanation is needed to explain why the proposed module works.

4) It is suggested to compare the proposed method with more recent state-of-the-art methods.

5) There are some grammatical or descriptive questions in the manuscript, which should be checked before submission.

Comments on the Quality of English Language

Moderate editing of English language required.

Author Response

Dear Reviewer,

Thank you very much for your valuable suggestions. We have made the necessary revisions. Please see the attachment for the detailed changes.

Best regards,

Xin Chi

Reviewer 2 Report

Comments and Suggestions for Authors

The manuscript proposes an innovative model EP-YOLOv8 network based on optical satellite images for the electric pylon detection. The model integrates LSK Block and DS-Conv in the SPPF, so that the model can better capture the detailed information. The model also adds EMS-Conv to the Head part to improve the feature expression ability. Finally, this manuscript emphasizes the improvement of EP-YOLOV8 network on the accuracy of electric pylon detection, and discusses its limitations and future research directions. Overall, there are some comments which may help to improve the quality of the manuscript.

1.    Although the authors propose an improved method for electric pylon detection, the description of the characteristics of electric pylons and the difficulties of current electric pylon detection is not very sufficient, which makes the problem not explored deeply enough. And the method proposed in this manuscript is not strongly targeted to electric pylon.

2.    Most of the methods introduced in this manuscript are existing methods, which are not obviously related to the characteristics of the electric pylons. The mechanism is not explored deeply enough, and there is a lack of certain innovation in terms of motivation. The authors should emphasize more the uniqueness of this manuscript’s method and highlight the connection between this manuscript’s innovation and the target characteristics of the electric pylon.

3.    There are some grammatical and logical errors in the manuscript.

4.    Lines 118 to 119 are repeated.

5.    The authors can make some adjustments to the content of the manuscript to make the order of innovations consistent.

6.    Please combine the dataset used in this manuscript to analyze the impact of the GSD of the remote sensing images used on the characteristics and detection of electric pylon. And combine these influences with the innovation of this manuscript to make the manuscript’s argument more convincing.

7.    The manuscript has not adequately introduced the data indicators that have an impact on electric pylon detection, which may affect the experimental results.

8.    Combining the results in Table 2 and Table 3, there seems to be some inconsistency in the results.

9.     In terms of accuracy, there is a lack of comparison with similar algorithms. Only the basic yolov8 is compared. Only the speed is compared with similar algorithms. The most important thing is that there is no comparison with similar electric pylon detection algorithms, making the effectiveness of this article questionable. Authors should add some comparative experiments with similar algorithms, and compare the experimental results from all aspects to better prove the effectiveness of the proposed method.

10.  What is the baseline in the “Ablation Experiment” part?

Author Response

(The authors gave the same response as above.)

Round 2

Reviewer 1 Report

Comments and Suggestions for Authors

I'm satisfied with the current version.

Comments on the Quality of English Language

The Quality of English Language is fine.

Reviewer 2 Report

Comments and Suggestions for Authors

No further comments.